# Determinants of Vaccine Hesitancy among Home Health Care Service Recipients in Saudi Arabia

**DOI:** 10.3390/vaccines11091436

**Published:** 2023-08-31

**Authors:** Walid A. Alkeridy, Hisham Alquaydheb, Shadan Almuhaidib, Nawal A. Sindi, Arwa Aljasser, Janet Kushner Kow, Amani S. Alqahtani

**Affiliations:** 1Department of Medicine, College of Medicine, King Saud University, Riyadh 12372, Saudi Arabia; 2Department of Medicine, Geriatric Division, University of British Columbia, Vancouver, BC V6T 1Z4, Canada; jkow@providencehealth.bc.ca; 3General Administration of Home Health Care, Therapeutic Affairs Deputyship, Riyadh 12382, Saudi Arabia; nawals@moh.gov.sa; 4Department of Internal Medicine, College of Medicine, Prince Sattam Bin Abdulaziz University, Al-Kharj 11942, Saudi Arabia; h.alquaydheb@psau.edu.sa; 5Scientific Research Center, Ministry of Defense Health Services, Riyadh 12426, Saudi Arabia; shadanmuhaidib@gmail.com; 6Department of Rehabilitation Sciences, College of Applied Medical Sciences, King Saud University, Riyadh 12372, Saudi Arabia; aljasser@ksu.edu.sa; 7Saudi Food and Drug Authority, Riyadh 13513, Saudi Arabia; as.qahtani@sfda.gov.sa

**Keywords:** vaccine hesitancy, home health care, Saudi Arabia

## Abstract

Background: Vaccine hesitancy has been identified by the World Health Organization (WHO) as a major worldwide health threat. Home Health Care (HHC) service recipients represent a vulnerable group and were prioritized to receive coronavirus disease (COVID-19) vaccination during the national vaccine campaigns in Saudi Arabia. We aimed to investigate the most frequent reasons for vaccine hesitancy among home health care recipients in Saudi Arabia. Methods: This cross-sectional survey was conducted among home health care (HHC) service recipients in Saudi Arabia from February 2022 to September 2022. The behavioral and social drivers (BeSD) model developed by the WHO was used to understand the factors affecting vaccination decision making in our cohort. Results: Of the 426 HHC service recipients enrolled in the study, a third were hesitant to complete the COVID-19 vaccination series. The most prevalent reported reason for COVID-19 vaccine refusal was concerns about the vaccine side effects (41.6%). Factors independently associated with COVID-19 vaccination hesitancy were: having chronic conditions (odds ratio [OR] = 2.59; 95% confidence interval [CI] = 1.33–5.05, *p* = 0.005), previous COVID-19 diagnosis (OR = 0.48; 95% CI: 0.28–0.82, *p* = 0.008), ease of getting the COVID-19 vaccine by themselves (OR = 0.49; 95% CI: 0.28–0.89, *p* = 0.018), belief in the importance of COVID-19 vaccine in protecting their health (OR = 0.60; 95% CI: 0.38–0.96, *p* = 0.032), and confidence in the safety of COVID-19 vaccination (OR = 0.38; 95% CI: 0.21–0.69, *p* = 0.001). Conclusion: Only one-third of the study participants were hesitant to complete the series of COVID-19 vaccination. Understanding the factors underpinning vaccine hesitancy among this group would help healthcare workers and policymakers in developing personalized health awareness campaigns aimed at improving vaccine acceptance levels.

## 1. Introduction

Vaccine hesitancy has been identified by the World Health Organization (WHO) as a major worldwide health threat [1]. Vaccine hesitancy is defined as a delay or refusal to take the recommended vaccine despite the vaccine’s availability [2]. As confidence in vaccine uptake declines, a myriad of negative health consequences ensue, including local outbreaks and financial and social disarray [3]. The success of any vaccination campaign is dependent on people’s confidence in both the public health authorities and the vaccines offered [4]. Home Health Care (HHC) service recipients represent a vulnerable group and were prioritized to receive COVID-19 vaccination during the national vaccine campaigns in Saudi Arabia. However, many public health reports in Saudi Arabia showed a slow uptake of the second dose of the COVID-19 vaccine among the HHC service recipients despite the vaccine’s availability. The factors influencing COVID-19 vaccination acceptance and skepticism in Saudi HHC service recipients are still unknown. The majority of HHC service recipients in Saudi Arabia are either older adults or people with disabilities (PWDs). Older adults are at an increased risk of complications from COVID-19 as well as hospitalization, critical care admission, functional decline, and death [5,6,7]. Recently, the reasons for vaccine hesitancy in low- and middle-income countries (LMICs) were investigated [8], showing an average acceptance rate among LMICs of 80.3%, with the lowest acceptance rates in Burkina Faso (66.5%) and Pakistan (66.5%); additionally, every sample from LMICs had a higher acceptance rate than the samples from the United States (64.6%) [8]. Interestingly, the most common reasons of vaccine hesitancy were related to concerns about the vaccine’s side effects.

This study aimed to explore the frequency of COVID-19 vaccine hesitancy among HHC service recipients as well as the study participants’ knowledge, attitude, and beliefs about COVID-19 vaccination. Moreover, we sought to examine the possible factors associated with vaccine hesitancy among HHC service recipients in Saudi Arabia.

## 2. Methods

### 2.1. Study Design and Population

A cross-sectional survey was conducted among patients receiving HHC services in Saudi Arabia between February 2022 and September 2022. The study was carried out in five main regions in Saudi Arabia, including the central, western, eastern, northern, and southern regions. We included participants enrolled in HHC services in the Ministry of Health (MOH) who were hesitant to complete the series of COVID-19 vaccine. We excluded those who were not receiving HHC services at the time of study and those who completed the series of COVID-19 vaccine. The Human Research Ethics Committee at the Ministry of Health reviewed and approved this study (Approval No: 21-104 M).

### 2.2. Sample Size

A successive sampling plan was used to ensure that we selected a sample that appropriately represented the diversity and characteristics of HHC service recipients in Saudi Arabia. Our goal was to include a diverse group of participants receiving home health care services from various geographical areas and demographics. The sample size was estimated with a confidence level of 95%, with the real value being within 5% of the measured/surveyed value; consequently, a sample size of 385 was regarded as adequate for this survey.

### 2.3. Questionnaire Development and Measurements

There are several well-validated measures for assessing vaccine hesitancy, including the Vaccine Hesitancy Scale (VHS). The VHS was developed to determine the factors contributing to vaccine hesitancy and has been widely used since its development in 2015 [9,10,11]. In this study, we utilized the behavioral and social drivers (BeSD) model to understand the factors influencing vaccination decision making among individuals [12,13,14,15,16]. In 2019, the BeSD working group was established by the WHO for the development and validation of an assessment tool. The development process encompassed multiple phases: (a) initial tool development, (b) field testing, (c) psychometric validation, and (d) finalization and selection of indicators [16]. During the field-testing phase, the working group successfully translated and linguistically validated the tool into various languages from multiple countries. Detailed information about the development process has been previously published [16]. The BeSD tool consists of four domains that influence vaccine uptake, including people’s perception and feeling about vaccination, social processes that drive or prevent vaccination, individual hesitancy to seek vaccination, and practical factors involved in seeking and receiving vaccination. Therefore, the survey forms included the following sections: (I) Socio-demographic characteristics; (II) General vaccination uptake status; (III) COVID-19 risk perception, vaccine stigma, and trust; (IV) COVID-19 vaccine—decision autonomy and reasons; (V) COVID-19 vaccine—family, community, and social norm; (VI) COVID-19 vaccine—confidence in providers; and (VII) COVID-19 vaccine information and information access.

The survey was piloted with the target population to ensure the questions flow well and are well-understood in the local context. The survey’s final version was also piloted with the targeted population (5%) to ensure that all recruitment processes designed by the researchers are feasible.

### 2.4. Recruitment Process

The survey was developed using a web-based questionnaire, integrated with a recruitment system and secured database to store the collected data. Participants were recruited through their phone numbers, which were obtained from the HHC patients’ database in Saudi Arabia’s MOH. After stratifying the numbers based on regions, trained interviewers contacted potential participants through their phone numbers to complete the questionnaire via a telephone-based interview. Informed consent for this study was obtained verbally over the phone from all participants prior to study recruitment. The study’s purpose, procedures, and the right to withdraw at any time were explained to the participants. Each participant was given three call attempts before being dropped from the list.

### 2.5. Statistical Analysis

The Statistical Package for Social Sciences version 29.0 (IBM Corp., Armonk, NY, USA) was used for data analyses. All tests were two-sided, and a *p*-value of ≤0.05 was considered statistically significant. The Shapiro–Wilks test was used to assess the normality of age distribution, which was found to be not normally distributed. Therefore, the median and interquartile range (IQR) were expressed as a measure of central tendency and compared using the non-parametric Mann–Whitney U test. Categorical variables were presented as frequencies and percentages, and compared using Pearson’s chi-squared test. Binary logistic regression analysis was used to explore the factors associated with the hesitancy of COVID-19 vaccination based on possible clinical significance. The responses to scale questions were transformed into dichotomous data to better present the results and simplify the message of this study. The responses with the words (moderate, a lot, very, or easy) were recoded into (yes), whereas those with the words (mild, a bit, or not at all) were recoded into (no). The factors that were found to be statistically significantly associated in the univariable analyses were included in the multivariable regression analyses. The results were reported as odds ratios (ORs) with the corresponding 95% confidence intervals (CIs).

## 3. Results

The study included 426 participants, of whom 227 (53.3%) were men. The median (IQR) of age was 75 (13) years, and almost 80% were aged ≥ 60 years. Three quarters of the study participants had chronic conditions. In particular, the most prevalent type of chronic condition was diabetes mellitus with a rate of 41.6%, followed by high blood pressure with a rate of 20.2%. The participants’ demographic characteristics are presented in Table 1. Most of the participants had not been previously diagnosed with COVID-19 (73%). Approximately one-third of the HHC participants were hesitant to complete the COVID-19 vaccination series. The participants’ knowledge, attitudes, and beliefs about the COVID-19 vaccine are shown in Table 2. The reasons for accepting or rejecting the COVID-19 vaccine are shown in Figure 1 and Figure 2. Two-thirds of the participants accepted the COVID-19 vaccine; 80.3% of the participants reported that they accepted the vaccine because they recognized its efficacy in protecting themselves from COVID-19, whereas only 8.1% accepted it because it was nationally mandated. Among participants who were hesitant to receive the vaccine, 41.6% reported that they refused to complete the COVID-19 vaccination series because they were concerned about the vaccine’s side effects.

Table 3 presents the final multivariable logistic regression analyses for factors associated with COVID-19 vaccine hesitancy. After adjusting for all statistically significant covariates, the factors associated with COVID-19 vaccine hesitancy were: having chronic conditions (OR = 2.59; 95% CI = 1.33–5.05), previous COVID-19 diagnosis (OR = 0.48; 95% CI: 0.28–0.82), ease of receiving the COVID-19 vaccine by themselves (OR = 0.49; 95% CI: 0.28–0.89), belief in the importance of COVID-19 vaccine in protecting their health (OR = 0.60; 95% CI: 0.38–0.96), and confidence in the safety of COVID-19 vaccine (OR = 0.38; 95% CI: 0.21–0.69). All other covariates were not statistically significantly related to the vaccine hesitancy.

## 4. Discussion

In the present cross-sectional study, 33.3% (142) of the participants did not want to complete the COVID-19 vaccine series despite the vaccine’s availability and widespread public campaigns encouraging high-risk groups to complete the vaccination series. The frequency of vaccine hesitancy in our cohort is close to that reported in North American studies [17]. The most common reason for refusing to complete the COVID-19 series is a concern about the vaccine’s side effects with a rate of 41.6% (59), similar to the findings of a previous study [8], followed by the belief that being homebound limits the risk of getting COVID-19 in 9.9% (14) of the study participants.

### 4.1. Vaccine Hesitancy and Chronic Conditions

Interestingly, in the multivariable model, participants with chronic conditions were more likely to be hesitant to receive vaccination (OR = 2.59; 95% CI: 1.33–5.05), *p* = 0.005). The most reported chronic condition was diabetes with a rate of 41.6% (134), followed by hypertension with a rate of 20.2% (65). Diabetes mellitus (DM) is associated with impaired immune system activation and reaction to severe acute respiratory syndrome coronavirus 2 (SARS-CoV-2) invasion [18]. DM leads to regulatory T lymphocyte dysfunction and cytokine hyperactivation [18]. Additionally, DM is well-known to increase the risk of developing severe SARS-CoV-2 infection as well as the risk of hospitalization, intensive care unit admission, and mortality [19,20,21]. HHC service recipients considered their chronic conditions as a possible contraindication to receive COVID-19 vaccination, despite the deliberate public health campaigns addressing the prioritization of vulnerable groups to receive COVID-19 vaccination. One explanation is the possibility that public messaging does not effectively reach older adults who are more likely to be illiterate and do not regularly use smartphones or social media, where most of the public health campaigns are announced [22,23,24].

### 4.2. Vaccine Hesitancy and Sociodemographic Factors

In concordance with previously published studies, we found that sex was associated with vaccine hesitancy [25,26,27,28,29,30]. In the univariable analysis, females were more likely than males to be vaccine-hesitant (OR = 1.71; 95% CI: 1.14–2.57, *p* = 0.009). However, this was not the case in the multivariate model. Historically, women are more likely to be hesitant to receive vaccines due to concerns regarding infertility and overall health outlook [31]. However, the majority of the current study participants were older women beyond the childbearing age; therefore, this explanation might not justify their attitude toward vaccine acceptance.

On the other hand, we found that among those who rejected the vaccine, only 32.4% made the decision on their own, whereas the majority relied on their families to make the decision regarding the vaccination. Family influence plays a significant role in healthcare decisions for Saudi people [32]. The nexus between personal agency and traditions in vaccine decision making is not simple, and reflects a complex interaction between familial background, culture, and personal outlook towards autonomy [32,33]. Given these sociodemographic factors, vaccine awareness campaigns should take into consideration the impact of family influence and collective decision making on vaccine acceptance.

### 4.3. The Association of Vaccine Knowledge, Accessibility, and Safety with Vaccine Hesitancy

In this study, a significant association was found between vaccine accessibility and vaccine hesitancy (Table 2). The majority of the study participants knew where to obtain the vaccine and felt that it was easy to obtain the vaccine (Table 2). Despite the hesitancy rate, most of the study participants preferred to receive the vaccination at home rather than going to vaccination centers. Indeed, early on in the COVID-19 pandemic, the Saudi MOH extended the offer of administering COVID-19 vaccinations at home to recipients of HCC services and older Saudi individuals nationwide.

In the context of perceptions regarding vaccine safety, the majority of study participants believed that COVID-19 vaccination was safe. However, we identified a discrepancy concerning the perceived importance of the vaccine in self-protection between those who accepted the vaccine and those who were hesitant to receive it (59.5% and 38%, respectively; see Table 2). However, most of the participants who were hesitant about receiving the vaccine were concerned about the vaccine’s side effects (41.6%) (Figure 2). This finding is consistent with the result of a recent systematic review that investigated the psychological factors contributing to COVID-19 vaccine hesitancy, which found that worries about vaccine safety and side effects are great contributors to vaccine hesitancy [34].

### 4.4. Future Directions

To mitigate the factors contributing to vaccine hesitancy in HHC, it is important for policy makers to develop health awareness campaigns that are personalized to older adults and PWDs from different backgrounds and capabilities who represent a heterogeneous group [35,36,37]. It is important to take into account the effect of disability- and age-related changes on the individual’s current function and abilities to interact with the targeted health awareness campaigns [36,37,38]. Therefore, effective public health messaging addressing vaccination acceptance and hesitancy in older adults and PWDs should be persuasive and clear, and address their cultural, socioeconomic, physical, and mental health needs [39,40,41]. On the other hand, it is noteworthy that, in Saudi Arabia, there is a sizable proportion of HHC service recipients who prefer to defer their decision about vaccination choices to their family; therefore, the vaccination health awareness campaigns need to address the factors resulting in the caregivers’ or substitute decision makers’ hesitancy to accept and pursue recommended vaccination which has proven efficacy and safety for the people they are trusted to look after [42]. The finding related to the presence of chronic disease as a predictor of vaccine hesitancy in HHC service recipients should not be overlooked, given the association of COVID-19 with worsened clinical outcomes including hospitalization and mortality, especially in people with chronic conditions [7]. It is well-established that hospitalization is associated with worsening functional, physical, and psychological outcomes in older adults [43,44,45]. HHC service recipients as a cohort represent a high-risk group for developing severe COVID-19 and poor clinical outcomes. Therefore, addressing these meaningful outcomes might resonate well with HHC service recipients and those providing care for them.

### 4.5. Strengths and Limitations

To the best of our knowledge, this is the first multi-regional study addressing the factors associated with COVID-19 vaccine hesitancy in HHC service recipients in Saudi Arabia. Despite this, some limitations should still be addressed. First, the sample size represents a convenient sample of long-term HHC service recipients, which limits the power of the study. Second, given the study’s observational nature, no causal inference could be gained; instead, significant associations were found, which require further studies to better understand these findings. Moreover, we did not collect information about educational level; therefore, we could not adjust for it in the multivariate model. Finally, frailty, which is a common syndrome in older adults, especially HHC service recipients, was not assessed in this survey-based study. Future studies could use validated questions to screen and adjust for frailty, which could potentially play a significant role in vaccine hesitancy among HHC service recipients.

## 5. Conclusions

In summary, this study found that one-third of HCC service recipients were hesitant to receive the COVID-19 vaccine. Concerns about the side effects was the most predominant reason for vaccine hesitancy.

Our study could help policymakers and healthcare providers in developing personalized health awareness campaigns aimed at improving the vaccine acceptance rate.

## Figures and Tables

**Figure 1 vaccines-11-01436-f001:**
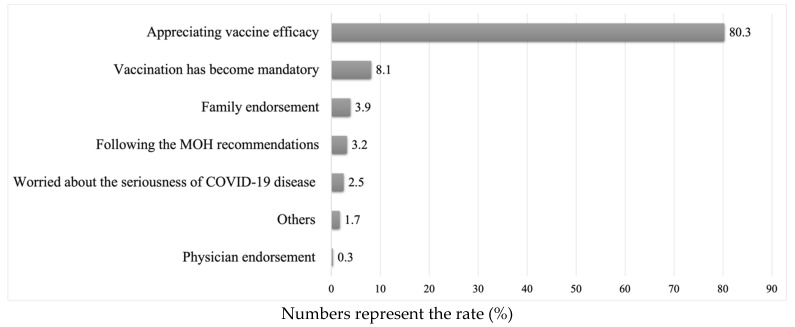
Reasons for accepting the COVID-19 vaccine (*n* = 284). Abbreviation: MOH Ministry of Health.

**Figure 2 vaccines-11-01436-f002:**
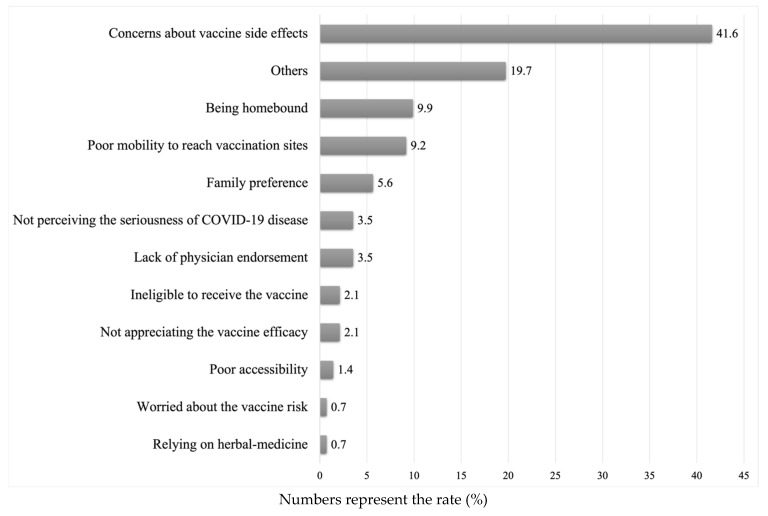
Reasons for rejecting the COVID-19 vaccine (*n* = 142).

**Table 1 vaccines-11-01436-t001:** Demographic characteristics of the study participants (*n* = 426).

Characteristics	N (%)
Age in years (Not normally distrusted *p* < 0.001)	
Median (IQR)	75 (13)
Gender	
Female	199 (46.7)
Male	227 (53.3)
Region	
Central	85 (20)
West	166 (39)
North	10 (2.3)
East	27 (6.3)
South	29 (6.8)
Not mentioned	109 (25.6)
Have chronic conditions	
Yes	322 (75.6)
Diabetes	134 (41.6)
Hypertension	65 (20.2)
Cardiovascular disease	25 (7.8)
Chronic kidney disease	5 (1.6)
Lung disorder	7 (2.2)
Dyslipidemia	27 (8.4)
Dementia	23 (7.1)
Cancer	4 (1.2)
Asthma	9 (2.8)
Immune disorder	2 (0.6)
Other conditions	21 (6.5)
Do not have any chronic condition	104 (24.4)
Diagnosed with COVID-19	
Yes	115 (27)
No	311 (73)
Received COVID-19 Vaccine	
Yes	284 (66.7)
No	142 (33.3)

Abbreviations: IQR: interquartile range, N: number of participants, *p*: *p*-value.

**Table 2 vaccines-11-01436-t002:** Participants’ knowledge, attitude, and beliefs about the COVID-19 vaccine (*n* = 426).

Factors	Vaccine-Hesitant	Vaccine-Accepting	*p*
	N (%)	N (%)	
Previous Vaccination History			
Ever received any vaccine as an adult			**<0.001**
Yes	12 (8.5)	208 (73.2)	
No	126 (88.7)	70 (24.6)	
Not sure	4 (2.8)	6 (2.1)	
Ever received seasonal vaccine			0.363
Yes	30 (21.1)	62 (21.8)	
No	102 (71.8)	211 (74.3)	
Not sure	10 (7)	11 (3.9)	
Knowledge about COVID-19 vaccine and role of decision making			
Know where to get COVID-19 vaccine			0.793
Yes	116 (81.7)	229 (80.6)	
No	26 (18.3)	55 (19.4)	
Preferred location to get COVID-19 vaccine			**<0.001**
Homecare teams	135 (95.1)	158 (55.6)	
Vaccine centers	3 (2.1)	118 (41.5)	
Primary care centers	3 (2.1)	2 (0.7)	
No preference	1 (0.7)	6 (2.1)	
In general, how easy is it to get COVID-19 vaccine			**0.018**
Easy	96 (67.6)	222 (78.2)	
Moderate	13 (9.2)	27 (9.5)	
Mild	7 (4.9)	12 (4.2)	
Not easy at all	26 (18.3)	23 (8.1)	
COVID-19 vaccine receipt decision making			**<0.001**
Me	46 (32.4)	188 (66.2)	
My son	37 (26.1)	49 (17.3)	
My daughter	11 (7.7)	16 (5.6)	
Brother	1 (0.7)	-	
Family consultation	16 (11.3)	3 (1.1)	
Grandson	-	1 (0.4)	
Husband	4 (2.8)	6 (2.1)	
Father	1 (0.7)	3 (1.1)	
Mother	3 (2.1)	1 (0.4)	
Treating physician	-	1 (0.4)	
Other people	23 (16.2)	16 (5.6)	
Confidence in COVID-19 vaccine and perception of COVID-19 disease			
Concerned about getting infected with COVID-19 virus			0.088
Not worried at all	68 (47.9)	137 (48.2)	
Bit worried	28 (19.7)	63 (22.2)	
Moderately worried	15 (10.6)	46 (16.2)	
Worried a lot	31 (21.8)	38 (13.4)	
Confidence in COVID-19 vaccine			**<0.001**
Do not trust it at all	28 (19.7)	12 (4.2)	
Mild trust	19 (13.4)	34 (12)	
Moderate trust	27 (19)	106 (37.3)	
Trust it a lot	68 (47.9)	132 (46.5)	
Importance of COVID-19 vaccine in protecting your health			**<0.001**
Not important at all	25 (17.6)	81 (28.5)	
Mildly important	48 (33.8)	17 (6)	
Moderately important	15 (10.6)	17 (6)	
Very important	54 (38)	169 (59.5)	
Confidence in the safety of COVID-19 vaccine			**<0.001**
Not safe at all	37 (26.1)	16 (5.6)	
A bit safe	18 (12.7)	44 (10.3)	
Moderately safe	28 (19.7)	103 (36.3)	
Very safe	59 (41.5)	139 (48.9)	
General knowledge about COVID-19 vaccine and role of decision maker in taking the vaccine			
Heard bad news about COVID-19 vaccine			0.836
Yes	62 (43.7)	121 (42.6)	
No	80 (56.3)	163 (57.4)	
Confidence in the information about COVID-19 vaccine provided by the Ministry of Health			**0.014**
Do not trust it at all	5 (3.5)	3 (1.1)	
Mild trust	13 (9.2)	9 (3.2)	
Moderate trust	30 (21.1)	72 (25.4)	
Trust it a lot	94 (66.2)	200 (70.4)	

Significant values are in bold. Abbreviations: N: number of participants, *p: p*-value.

**Table 3 vaccines-11-01436-t003:** Covariates associated with COVID-19 vaccine hesitancy (*n* = 426).

Factors	Univariable Analysis	*p*	Multivariable Analysis	*p*
	Odd Ratio (95% CI)		Odd Ratio (95% CI)	
Demographic Characteristics				
Age		**<0.001**		0.072
≥60 years	4.35 (2.23–8.50)		2.02 (0.94–4.35)	
<60 years	reference		reference	
Gender		**0.009**		0.111
Female	1.71 (1.14–2.57)		1.45 (0.92–2.29)	
Male	reference		reference	
History				
Chronic conditions		**<0.001**		**0.005**
Yes	3.54 (1.99–6.30)		2.59 (1.33–5.05)	
No	reference		reference	
Diagnosed with COVID-19		**0.005**		**0.008**
Yes	0.49 (0.30–0.81)		0.48 (0.28–0.82)	
No	reference		reference	
Access to vaccination				
Ever received seasonal vaccine?		0.373	-	-
Yes	reference			
No	1.00 (0.61–1.64)			
Not sure	1.88 (0.77–4.57)			
Know where to get COVID-19 vaccine		0.793	-	-
Yes	1.07 (0.64–1.80)			
No	reference			
In general, how easy is it to get COVID-19 vaccine by yourself?		**0.004**		**0.018**
Easy	0.46 (0.27–0.79)		0.49 (0.28–0.89)	
Not easy	reference		reference	
Confidence in and perceived safety and importance of COVID-19 vaccine				
Worried about getting infected with COVID-19 virus		0.552	-	-
Worried	1.14 (0.74–1.76)			
Not Worried	reference			
Confidence in COVID-19 vaccine		**<0.001**		0.713
Trust	0.39 (0.24–0.63)		0.89 (0.47–1.68)	
Do not trust	reference		reference	
Importance of COVID-19 vaccine in protecting your health		**<0.001**		**0.032**
Important	0.50 (0.33–0.75)		0.60 (0.38–0.96)	
Not important	reference		reference	
Confidence in the safety of COVID-19 vaccine		**<0.001**		**0.001**
Safe	0.28 (0.17–0.44)		0.38 (0.21–0.69)	
Not safe	reference		reference	
Knowledge and level of trust in information provided by MOH about COVID-19 vaccine				
Confidence in the information about COVID-19 vaccine provided by the Ministry of Health		**0.002**		0.501
Trust	0.30 (0.14–0.65)		0.73 (0.30–1.81)	
Do not trust	reference		reference	
Heard bad about COVID-19 vaccine		0.84	-	-
Yes	1.04 (0.70–1.57)			
No	reference			

Significant values are in bold. Abbreviations: CI: confidence interval, OR: Odds ratio, *p*: *p*-value.

## Data Availability

Data are available upon reasonable request to the corresponding author.

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
