# Peer review of "Determinants of Vaccine Hesitancy among Home Health Care Service Recipients in Saudi Arabia"

_vaccines, 2023, doi:10.3390/vaccines11091436_

Round 1

Reviewer 1 Report

This is a well-written short article dealing with the COVID vaccine hesitancy of a sample of older chronic diseases patients attended by the health care system of Saudi Arabia. The authors adequately analyze the sample and most of the conclusions are valid, explained and discussed. There a paragraph that deserves a special consideration, from lines 167-180, related to chronic conditions and hesitancy. The discussion at this paragraph was not related to their statistical data, because there no significant data in table and the analysis could be biased by age group and literacy, as demography data also had several differences, thus their conclusion only reflects demographic effect. I suggest to the authors to focus on their significant data, avoiding discussions on chronic diseases and hesitancy. It is obvious that old people with illiteracy must rejected vaccines due to cultural background, as adequately discussed in the next paragraph.

It this speculative discussion is removed with minor modifications, the article could be published.

Author Response

I would like to extend my sincere gratitude to you for the meticulous review of my paper. your insightful comments and constructive feedback have been invaluable in enhancing the quality of this manuscript. Thank you once again for your expertise and dedication to improving this work.

With regards to your comment about the association of chronic conditions  with vaccine hesitancy, we would like to point to you that we adjust for age and gender in the multivariate analysis, however, we did not adjust for educational level, because the information about education was not available for a majority of participants, therefore, we could not adjust for it in the multivariate model. However, we added it as a limitation in our study. We still think it is important to address this point because there is significant misinformation and widespread false news in the social media that people with chronic conditions should avoid vaccination as it might complicate their health conditions.

Reviewer 2 Report

Dear Author,

Greetings!

1  Home Health Care (HHC) service recipients represent 21

a vulnerable group and were prioritized to receive coronavirus disease (COVID-19) vaccination 22

during the national vaccine campaigns in Saudi Arabia. We aimed to investigate the most frequent 23

reasons for vaccine hesitancy among older Saudis. Methods: This cross-sectional survey was con- 24

ducted among home health care (HHC) service recipients in Saudi Arabia from February 2022 to 25

September 2022. The behavioral and social drivers (BeSD) model developed by the WHO was used 26

to understand the factors affecting vaccination decision making in our cohort.(PLEASE REVISE THIS AND ADD UPDATED ONE)

2The most prevalent reported reason for COVID-19 vaccine refusal was worrying 29

about the side effects (41.6%). Factors independently associated with COVID-19 vaccination hesi- 30

tancy were having chronic conditions (odds ratio [OR] = 2.59; 95% confidence interval [CI] = 1.33– 31

5.05, p = 0.005), previous COVID-19 diagnosis (OR = 0.48; 95% CI: 0.28–0.82, p = 0.008), ease of getting 32

the COVID-19 vaccine by themselves (OR = 0.49; 95% CI: 0.28–0.89, p = 0.018), belief of the im- 33

portance of COVID-19 vaccine in protecting their health (OR = 0.60; 95% CI: 0.38–0.96, p = 0.032), 34

and confidence in the safety of COVID-19 vaccination (OR = 0.38; 95% CI: 0.21–0.69, p = 0.001).(pLEASE CHECK THIS AGAIN)

3Study design and population 69

A cross-sectional survey was conducted among patients receiving HHC services in 70

Saudi Arabia between February 2022 and September 2022. The study was carried out in 71

five main regions in Saudi Arabia, including the central, western, eastern, northern, and 72

southern regions. We included participants enrolled in HHC service in the MOH and were 73

hesitant to complete the series of COVID-19 vaccine. We excluded those who were not 74

receiving HHC service at the time of study and those who completed the series of COVID-(PLEASE MODIFY AS PER OUR JOURNAL GUIDE LINES)

4 Please explain clearly sample size A consecutive sampling plan was used to ensure a sample that was representative of 79

HHC service recipients in Saudi Arabia. The sample size was calculated with a confidence 80

level of 95%, with the real value within ±5% of the measured/surveyed value; thus, a sam- 81

ple size of 385 was deemed sufficient for this survey

5 Tables were not clear please follow journal guide lines

6 Please check plagiarism by using turnit in 

7 References were not arranged properly please use zotero or mendeley

Best Regards 

Language minor corrections were required as per concern with English language .

Tab`le contents and units were not clear

Author Response

I would like to extend my sincere gratitude to you for the meticulous review of my paper. your insightful comments and constructive feedback have been invaluable in enhancing the quality of this manuscript. Thank you once again for your expertise and dedication to improving this work.

We apologize for having difficulty understanding some of the comments, perhaps due to formatting issues. However, we assure you that we did our best to address them best as we could.

Please find the responses to your comments in bold

4 Please explain clearly sample size A consecutive sampling plan was used to ensure a sample that was representative of 79 . 

HHC service recipients in Saudi Arabia. The sample size was calculated with a confidence 80

level of 95%, with the real value within ±5% of the measured/surveyed value; thus, a sam- 81

ple size of 385 was deemed sufficient for this survey. Addressed in the revised version 

5 Tables were not clear please follow journal guidelines. We revised the tables to address this comment

6 Please check plagiarism by using turnit in 

7 References were not arranged properly please use zotero or mendeley. We revised the references and used Endnote and accepted software per the journal guidelines 

Reviewer 3 Report

Dear Authors,

The topic of the manuscript is interesting and suitable for the Vaccines journal.

Still, there are some concerns from my point of view (especially the discussion part and some methodological aspects):

What is the novelty brought by this study?

Methods: there is a recently developed Multidimensional Vaccine Hesitancy Scale (MVHS). Howard, M.C. A more comprehensive measure of vaccine hesitancy: Creation of the Multidimensional Vaccine Hesitancy Scale (MVHS). J. Health Psychol. 2021, 27, 2402–2419. Shapiro, G.K.; Tatar, O.; Dube, E.; Amsel, R.; Knauper, B.; Naz, A.; Perez, S.; Rosberger, Z. The vaccine hesitancy scale: Psychometric properties and validation. Vaccine 2018, 36, 660–667. Perhaps the authors should mention this and add some references.

Page 1, the abstract, the aim (line 24): “among older Saudis”. Why only older people? The recipients are older and persons with disabilities (the authors mentioned this issue – lines 55-56). Perhaps the authors should add in the title “older recipients”.

Line “Vaccine hesitancy may be prevalent among HHC service recipients in Saudi Arabia” Please rephrase, only one-third of the subjects were hesitant.

Page 2, line 73: MOH, please mention the Ministry of Health

Page 3, lines 95-96: Please mention the reference for the linguistic validation process and add some details.

Page 3, line 105: Please mention the way the interviewers get the informed consent (oral?)

Line 168-169: the phrase is repetitive please remove it.

Line 181-191: please rephrase, it is not clear (Males or females are more hesitant?). Please get another explanation. Moreover, “Contrarily, among those who rejected the vaccine, only 32.4% made the decision on their own, whereas the majority relied on their families to make the decision regarding the vaccination” – how this aspect is related to gender?

Line 195: “Despite the hesitancy rate, most of the study participants preferred to receive the vaccination at home”. Please explain.

Lines 202-207: the explanation does not succeed to explain the results mentioned “Interestingly, among participants who were hesitant to receive the vaccine, 41.6% reported that they refused to complete the COVID-19 vaccination series because they were concerned about the vaccine’s side effects” (lines 137-139). Rather, “it was to be expected” than “Interestingly” as you also mentioned, “This finding is consistent with the result of a recent systematic review”.

Line 238: “frailty, which is a common syndrome in older adults, especially HHC service recipients, was not captured in this survey-based study”: What do you mean by captured? Please be clearer.

Conclusion: only policymakers should be involved? What about the health care providers (doctors and nurses) from home health care services?

In the text, the reference number should be written after the comma?

Author Response

I would like to extend my sincere gratitude to you for the meticulous review of my paper. your insightful comments and constructive feedback have been invaluable in enhancing the quality of this manuscript. Thank you once again for your expertise and dedication to improving this work.

please see the response your comments below in bold 

Methods: there is a recently developed Multidimensional Vaccine Hesitancy Scale (MVHS). Howard, M.C. A more comprehensive measure of vaccine hesitancy: Creation of the Multidimensional Vaccine Hesitancy Scale (MVHS). J. Health Psychol. 2021, 27, 2402–2419. Shapiro, G.K.; Tatar, O.; Dube, E.; Amsel, R.; Knauper, B.; Naz, A.; Perez, S.; Rosberger, Z. The vaccine hesitancy scale: Psychometric properties and validation. Vaccine 2018, 36, 660–667. Perhaps the authors should mention this and add some references. although we agree with this comment, the BeSD was rigorously validated and translated for several languages during development. (reference were added). however, we see the valid point and will include in the manuscript.

Page 1, the abstract, the aim (line 24): “among older Saudis”. Why only older people? The recipients are older and persons with disabilities (the authors mentioned this issue – lines 55-56). Perhaps the authors should add in the title “older recipients”. We agree and we addressed this aim and removed the word older from aim since our study included younger patients with disability.

Line “Vaccine hesitancy may be prevalent among HHC service recipients in Saudi Arabia” Please rephrase, only one-third of the subjects were hesitant. We agree and have revised it accordingly.

Page 2, line 73: MOH, please mention the Ministry of Health We agree and have revised it accordingly.

Page 3, lines 95-96: Please mention the reference for the linguistic validation process and add some details. We agree and have revised it accordingly. We explained further details and added a references.

Page 3, line 105: Please mention the way the interviewers get the informed consent (oral?) We agree and have revised it accordingly

Line 181-191: please rephrase, it is not clear (Males or females are more hesitant?). Please get another explanation. Moreover, “Contrarily, among those who rejected the vaccine, only 32.4% made the decision on their own, whereas the majority relied on their families to make the decision regarding the vaccination” – how this aspect is related to gender? We agree and have revised it accordingly.

“Despite the hesitancy rate, most of the study participants preferred to receive the vaccination at home”. Please explain. We agree and added few explanations.

Lines 202-207: the explanation does not succeed to explain the results mentioned “Interestingly, among participants who were hesitant to receive the vaccine, 41.6% reported that they refused to complete the COVID-19 vaccination series because they were concerned about the vaccine’s side effects” (lines 137-139). Rather, “it was to be expected” than “Interestingly” as you also mentioned, “This finding is consistent with the result of a recent systematic review”. We revised it accordingly. Rephrased the scetion be clearer

 “frailty, which is a common syndrome in older adults, especially HHC service recipients, was not captured in this survey-based study”: What do you mean by captured? Please be clearer. We revised it accordingly

Conclusion: only policymakers should be involved? What about the health care providers (doctors and nurses) from home health care services? We revised it accordingly

In the text, the reference number should be written after the comma? Revised it accordingly

Round 2

Reviewer 3 Report

Dear authors, my sugestions have been revised. 
Good luck!